

# Quantifying uncertainty in annual runoff due to missing data

Craig R. See[1,*], Mark B. Green[2,3,*], Ruth D. Yanai[4], Amey S. Bailey[3], John L. Campbell[3] and Jeremy Hayward[5,†]

[1] Department of Ecology, Evolution, and Behavior, University of Minnesota, St. Paul, MN, United States of America
[2] Department of Earth, Environmental, and Planetary Sciences, Case Western Reserve University, Cleveland, OH, United States of America
[3] Northern Research Station, USDA Forest Service, Durham, NH, United States of America
[4] Department of Sustainable Resources Management, State University of New York College of Environmental Science and Forestry, Syracuse, NY, United States of America
[5] Department of Environmental and Forest Biology, State University of New York College of Environmental Science and Forestry, Syracuse, NY, United States of America
* These authors contributed equally to this work.
† Deceased.

## ABSTRACT

Long-term streamflow datasets inevitably include gaps, which must be filled to allow estimates of runoff and ultimately catchment water budgets. Uncertainty introduced by filling gaps in discharge records is rarely, if ever, reported. We characterized the uncertainty due to streamflow gaps in a reference watershed at the Hubbard Brook Experimental Forest (HBEF) from 1996 to 2009 by simulating artificial gaps of varying duration and flow rate, with the objective of quantifying their contribution to uncertainty in annual streamflow. Gaps were filled using an ensemble of regressions relating discharge from nearby streams, and the predicted flow was compared to the actual flow. Differences between the predicted and actual runoff increased with both gap length and flow rate, averaging 2.8% of the runoff during the gap. At the HBEF, the sum of gaps averaged 22 days per year, with the lowest and highest annual uncertainties due to gaps ranging from 1.5 mm (95% confidence interval surrounding mean runoff) to 21.1 mm. As a percentage of annual runoff, uncertainty due to gap filling ranged from 0.2–2.1%, depending on the year. Uncertainty in annual runoff due to gaps was small at the HBEF, where infilling models are based on multiple similar catchments in close proximity to the catchment of interest. The method demonstrated here can be used to quantify uncertainty due to gaps in any long-term streamflow data set, regardless of the gap-filling model applied.

# INTRODUCTION

Accurately estimating stream runoff is essential to water and nutrient budgets in watershed studies, but long-term data sets inevitably contain gaps in their records. Missing or unusable streamflow data occur due to planned maintenance, equipment failure, and other disruption of stage measurements. For calculations requiring a continuous record, such

Corresponding author
Craig R. See, crsee@umn.edu

as annual runoff and solute export, infilling data gaps with best estimates is necessary. The use of infilled streamflow values adds uncertainty to runoff calculations, but this uncertainty is rarely reported in the literature, despite its importance to hydrologic budgets and estimates of solute export (*Lloyd et al., 2016*). It was recently demonstrated that uncertainties in discharge contribute heavily to estimates of phosphorus export in suspended solids (*Krueger et al., 2009*). Historically, streamflow gaps have often been filled using statistical models that assume that the data conform to probability distributions (e.g., *Dempster, Laird & Rubin, 1977*; *Simonovic, 1995*; *Rubin, 1996*), allowing for the calculation of parametric uncertainty estimates (e.g., prediction intervals). Increasingly, however, non-parametric methods of prediction are being used to fill streamflow gaps (e.g., *Zealand, Burn & Simonovic, 1999*; *Khalil, Panu & Lennox, 2001*; *Mwale, Adeloye & Rustum, 2012*). These methods require estimating uncertainty using numerical techniques.

The contribution of gaps in streamflow data to uncertainty in runoff has been quantified for national networks of gauged watersheds in Canada and the USA (*Kiang et al., 2013*; *Mishra & Coulibaly, 2010*). Less is known about data gaps in small watershed studies, though some evidence suggests the effects may be large enough to affect annual water budgets (*Campbell et al., 2016*). In a recent survey hydrologists ranked infilling gaps in the discharge record among the most important sources of uncertainty in streamflow monitoring (*Yanai, See & Campbell, 2018*). Methods reported for infilling streamflow gaps include manual inference ("eyeballing it;"*Rees, 2008*; *Yanai et al., 2015*), regression models based on predictive variables (*Beauchamp, Downing & Railsback, 1989*; *Elshorbagy, Panu & Simonovic, 2000*), process-based models (*Beven, 2012*; *Gyau-Boakye & Schultz, 1994*), and artificial neural networks (*Ilunga & Stephenson, 2005*; *Khalil, Panu & Lennox, 2001*). To date, studies addressing the issue of streamflow gaps have largely focused on comparing the efficacy of various infilling models in larger rivers (e.g., *Hamilton & Moore, 2012*; *Harvey, Dixon & Hannaford, 2012*), but few have attempted to quantify the uncertainty that these modeled values introduce into runoff estimates (*Campbell et al., 2016*; *Mwale, Adeloye & Rustum, 2012*).

In this paper, we describe the causes and duration of gaps in the discharge record for a reference stream at the Hubbard Brook Experimental Forest for the period 1996–2009. We demonstrate an approach to quantifying the uncertainty introduced by infilling data gaps and describe the effects of gap length and flow rate on the uncertainty introduced.

## METHODS

### Study site and dataset

The Hubbard Brook Experimental Forest, New Hampshire, USA, contains six small headwater catchments clustered on a south-facing slope (Table 1) that have been monitored for many decades (since at least 1963). Streams draining these catchments are routed through 90° or 120° V-notch weirs and, in two catchments, San Dimas flumes, to estimate discharge ($Q$) by measuring stage height in these hydraulic structures. From 1963 to 2012, stage heights were recorded using Leupold-Stevens A-35 strip chart recorders with 7-jewel Chelsea Marine clocks for the V-notches and Belfort FW-1 recorders for the

**Table 1  Characteristics of 6 south-facing watershed at the Hubbard Brook Experimental Forest.** Aspect, slope and elevation represent watershed means (range shown in parentheses). Analyses of uncertainty due to gaps were conducted on watershed 6, with watersheds 1–5 used as predictors in the infilling model.

| Watershed | Area (ha) | Aspect | Slope (°) | Elevation (m) |
|---|---|---|---|---|
| 1 | 11.8 | S 22 °C E | 19.7 | 623 (448–747) |
| 2 | 15.6 | S 31 °C E | 19.6 | 613 (503–716) |
| 3 | 42.4 | S 23 °C W | 17.1 | 632 (527–732) |
| 4 | 36.1 | S 40 °C E | 16.8 | 608 (442–747) |
| 5 | 21.9 | S 24 °C E | 17.5 | 635 (488–462) |
| 6 | 13.2 | S 32 °C E | 16.1 | 683 (549–792) |

flumes. Depending on $Q$, between 2 and 130 inflection points per day were manually digitized from the strip charts. When gaps occurred, technicians filled them on the strip charts by visually comparing them to the hydrographs from nearby streams. The digitized inflection points describing stage height on the strip charts were converted to $Q$ using the theoretical stage-discharge relationship (*Bailey et al., 2003*). A 5-minute record of discharge was generated based on linear interpolation between inflection points, which we will refer to as "observations."

## Characterizing Streamflow Gaps

We identified the causes and duration of all gaps from 1996 to 2009 in Watershed 6 (W6), a reference watershed at HBEF that has been the focus of extensive long-term ecological studies (*Likens, 2013*). Prior to 1996, the occurrence and causes of gaps were not documented. Using field notes, we categorized the cause of each gap as equipment failure, technician error (e.g., failing to replace strip chart paper or to tighten the pen on the chart recorder), planned maintenance, or unreliable data due to ice or debris in the weir (*Campbell et al., 2016*). We characterized $Q$ for each observation during the gap as described below.

## Uncertainty due to streamflow gaps
### Filling gaps in streamflow

We filled gaps in discharge from W6 using an ensemble of predictions based on the five other south-facing weirs (Table 1). Ensemble predictions are generally more robust and are more accurate than single models (*Ren, Zhang & Suganthan, 2016*). Five-minute observations from 1963 to 2012 (5,259,601 observations total) were used to model the individual relationships in specific discharge ($q_W$, where W identifies the predictor watershed) between watersheds. Because the relationship in discharge between W6 and the predictor watersheds is flow-dependent, we used a binned regression approach. These relationships were developed by binning 95 $q_W$ by percentile P ($q_{W,P}$) and calculating the median $q_{W,P}$ for each percentile ($\tilde{q}_{W,P}$), with the exception of the 1st and 99th percentiles, which are discussed below. We calculated the median $q$ in W6 ($\tilde{q}_{6,P_W}$) for the timestamps corresponding to the $q_{W,P}$ of each of the five predictor watersheds. The values of $\tilde{q}_{6,P_W}$ were linearly interpolated between percentiles (Fig. 1A). To avoid predicting negative flow,

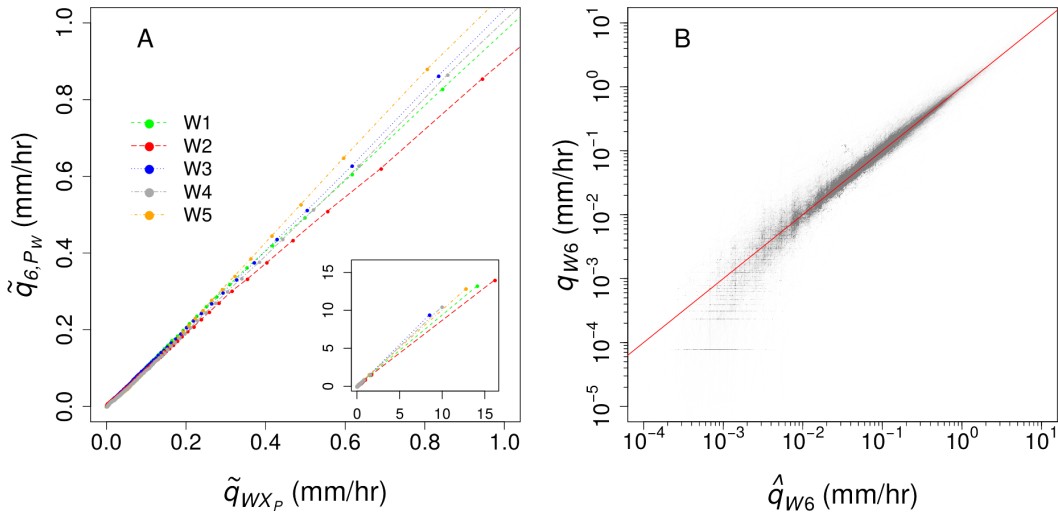

**Figure 1** (A) The relationships between specific discharge ($q$) in watershed 6 and the five predictor watersheds, with an inset showing the full range of recorded discharge; (B) the comparison of measured and ensemble-predicted 5-minute $q$ for the record of study ($n = 5,259,601$) compared to the 1:1 line (red).

$q_w$ values less than $q_{W,0.5}$ were linearly interpolated to (0,0). For $q_W > q_{W,99.5}$, we fit a least squares linear regression to paired $q_W$ and $q_6$ values. The resulting relationships produced five separate predictions for each observation of $q_6$ (Fig. 1A), and the median of these values was selected as the best prediction, denoted $\hat{q}_6$ (Fig. 1B). We used these predictions to estimate $q_6$ during real and simulated gaps in the record.

The $\hat{q}_6$ values were often offset from the actual values at the beginning or end of a gap. To correct this, we forced, or "snapped," the predicted flow to match the measured values at the beginning and end of each gap. This was done using the ratio of actual to predicted values at the beginning and end of the gap and multiplying by a linearly interpolated ratio for all $\hat{q}_6$ values during the gap (Fig. 2). Similar approaches have been used to fill gaps in streamwater chemical concentrations between samples using flow-concentration relationships (*Aulenbach & Hooper, 2006*; *Vanni et al., 2001*) and gaps in soil respiration data using relationships with temperature (*Bae et al., 2015*). The resulting model predicted $q_6$ with a Nash-Sutcliffe efficiency of 0.95 [Figure 1b;](*Nash & Sutcliffe, 1970*).

### Effects of gap length and flow rate on runoff uncertainty

To examine the effect of gap duration and hydrologic conditions on runoff ($RO$) uncertainty in W6, we simulated gaps of ten different durations ranging from one hour to one month (1, 6, 12, 24, 48, 96, 168, 336, 504, and 672 h). For each gap duration we randomly created 100,000 simulated gaps in the W6 record, excluding time periods that included real gaps. For each simulated gap, we calculated the error associated with gapfilling by subtracting the total $RO$ observed in the data from the total $RO$ predicted by our ensemble model. To describe the effect of $q$ on gap uncertainty we divided gaps of each duration into octiles

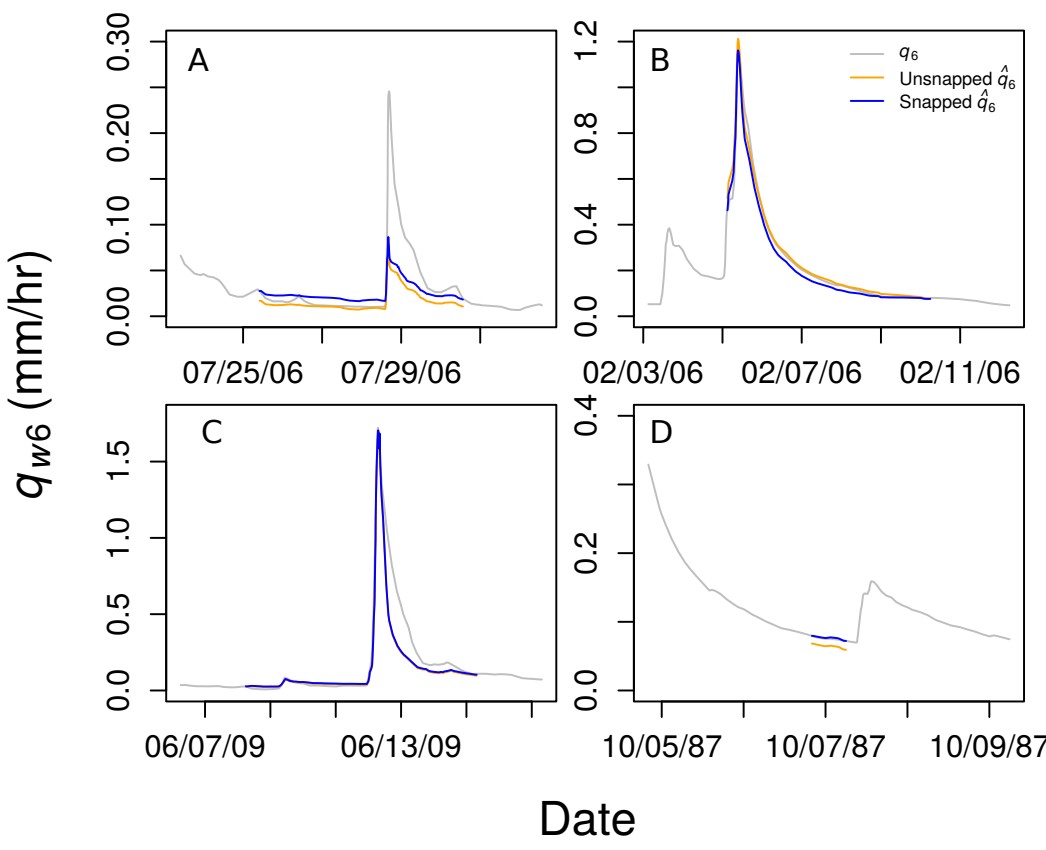

**Figure 2** **Four examples of the predicted discharge in Watershed 6 ($q_6$) from the ensemble regression ($\hat{q}_6$) compared to the actual $q_6$.** Snapped predictions are shown in blue and unsnapped predictions in orange. Snapping improved predictions sometimes (A and D) but not always (B and C).

based on the mean $q$ during each simulated gap, resulting in 12,500 simulated gaps for each combination of duration and flow-rate octile.

### Uncertainty in annual runoff due to gaps at Hubbard Brook

We applied a similar approach to that described above to calculate the uncertainty in annual $RO$ ($RO_{ann}$) due to gaps in the $q_6$ record. For each real gap ($G$) in the record, we randomly sampled from the entire record to create 10,000 simulated gaps ($G'$) of the same duration as the actual gap. To ensure similar hydrologic conditions between $G$ and $G'$, we calculated the simulated $RO$ over the period of $G$ ($RO_G$) and the measured $RO$ for each associated $G'$ ($RO_{G'}$), then eliminated the 90% of $RO_{G'}$ values with the largest absolute difference between $RO_G$ and $RO_{G'}$. Using the remaining 1000 $RO_{G'}$ values for each $G$, we calculated 1000 possible values of the error ($E_G$) between predicted runoff $\hat{RO}_{G'}$ and measured $RO_{G'}$ during the simulated gap ($E_G = \hat{RO}_{G'} - RO_{G'}$). To estimate uncertainty due to gaps in each water year, an estimate of the distribution of annual error ($E_{ann}$) was generated by iteratively summing $G'$ for each $G$ in the water year, over 1000 iterations.

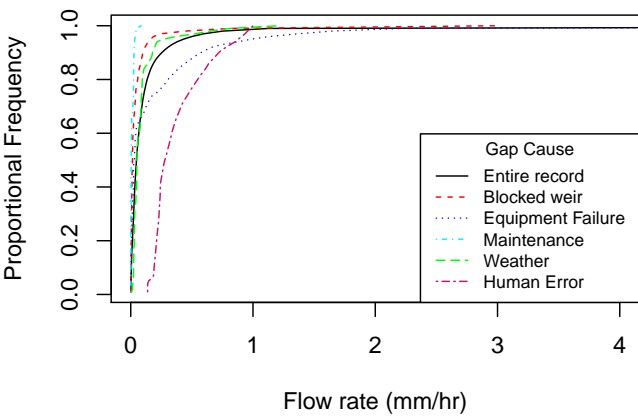

**Figure 3** Cumulative frequencies of predicted flow rates during gaps categorized by cause, along with the cumulative frequency of flow rates for all observations for the same time period (shown in black). Time period represents data from 1996–2009.

## RESULTS

All years we analyzed (1996–2009) contained gaps in the record. On average there were $21.9 \pm 8.5$ (s.d.) gaps per year, lasting an average of $1.1 \pm 0.3$ days. The median gap length in our dataset ($n = 270$) was 13.5 h. Individual gaps ranged in duration from 30 min to 12 days. The most common cause of gaps in terms of total time was debris (44% of the duration of all gaps) and ice (25% of total duration) in the v-notch. Equipment failure (mostly due to problems with chart recorder clocks) accounted for 21% of the gap time, while 7% was due to maintenance and repairs, and 3% due to human error (such as failing to correctly load or collect a chart or replace a pen).

The causes of gaps varied with $Q$ (Fig. 3). Maintenance is normally scheduled at low flow, and this is reflected in the distribution of $Q$ during gaps associated with maintenance. Debris in the weir was also responsible for gaps at lower than average $Q$, perhaps because higher $Q$ clears debris from the notch. Conversely, ice events and equipment failure caused gaps at higher than average $Q$. Ice blockages commonly occur during the high flow snowmelt period and equipment can be damaged during flood events throughout the year.

The uncertainty associated with $RO$ during individual gaps averaged 2.8% and ranged from 0–60% of $RO$ (95% CI based on 100,000 simulations). The uncertainty in runoff estimates during our simulated gaps was affected by both gap duration and $Q$ (Fig. 4). Short gaps applied during low $Q$ had the smallest differences between simulated and observed runoff. During gaps with low average $Q$ (ca. $< 1$ l/s), gap duration appeared to be the primary driver of uncertainty, while $Q$ became more important at higher average $Q$ (Fig. 4). We found that runoff during the gap–the product of gap duration and flow rate–was a strong predictor of the uncertainty associated with the gap. For 80 simulated gaps of different durations and flow rates, the standard deviation of the 1,000 estimates of error during the gap ($E_G$) was a linear function of estimated runoff during the gap (Fig. 5).

Gaps contributed surprisingly little uncertainty to estimates of annual runoff at Hubbard Brook, which averaged 866 mm/year for W6 from 1996 to 2009. The annual range in

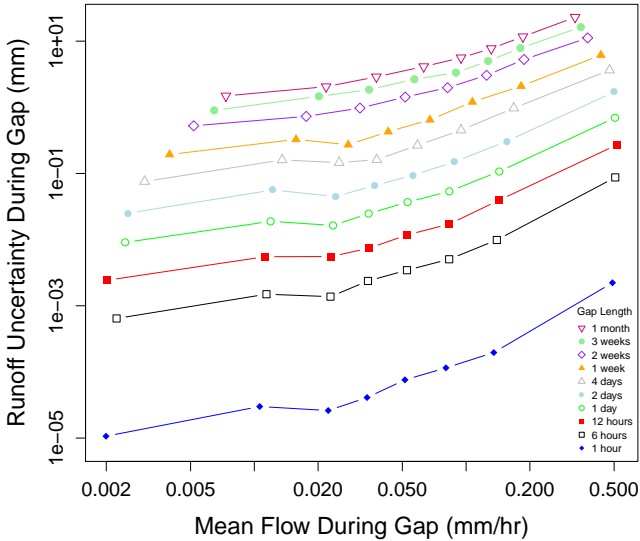

**Figure 4** **The uncertainty in predicted *RO* during a gap increases with gap duration and *q* during the gap.** Results of 10,000 simulated streamflow gaps for each of 10 gap lengths. Open circles represent the median values for octiles based on runoff ($n = 1,250$ simulations).

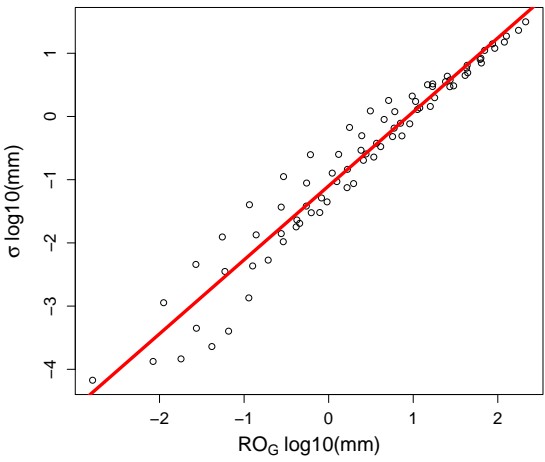

**Figure 5** **Uncertainty (calculated as the standard deviation, $\sigma$, in log10 watershed mm) for 80 artificially-created streamflow gaps as a function of the total runoff during each gap.** Predictions are based on an ensemble regression of 5 adjacent small catchments. The red line is the best-fit ordinary least-squares regression: $log10(\sigma) = 1.17(log10(RO_G)) - 1.1$.

uncertainty due to filling gaps averaged 11 mm/year (95% CI), but this varied considerably by year. Of the years included in our analysis, the annual uncertainty due to gaps was highest in 2009, with an estimated runoff of 1,194 mm and 95% confidence between 1,181 and 1,212 mm (Table 2). Uncertainty was lowest in 2006, with an estimated runoff of 1031 mm and a 95% CI of 1,030 to 1,032 mm. When considered as a percentage, the year with

the highest annual uncertainty was again 2009, with a 95% CI of 2.7%, and the year with lowest uncertainty was 2006, with a 95% CI representing 0.2% of annual runoff.

## DISCUSSION

When all gap types were considered together, gaps occurred disproportionately during low flow, but this trend may not hold true in larger watersheds. In an analysis of nationally archived river flow data in the UK, data gaps tend to occur during high-flow events (*Marsh, 2002*). This may be due to differences in equipment. Gauging stations for larger streams and rivers do not use v-notch weirs, and may not have the same issues associated with debris obstructions, which tend to occur during low flow at Hubbard Brook (Fig. 3).

Uncertainty in discharge during gaps was orders of magnitude higher for longer gaps and at higher flow (Fig. 4), with runoff during the gap being a better predictor than either gap duration or $Q$ alone (Fig. 5). The errors presented here generally follow highly skewed distributions (note the logarithmic axes in Figs. 4 and 5), clearly highlighting that the largest contributors to runoff uncertainty are the gaps occurring at high flows. In our case study, with multiple gauged streams in close proximity to one another, it was perhaps not surprising that an ensemble of regression relationships predicted runoff with high accuracy. Uncertainty would likely be greater when infilling gaps in larger, unpaired watersheds (e.g., *Kiang et al., 2013*), though it was recently demonstrated that when <10% of the annual record was missing, calibrated hydrologic models performed quite well across a wide range of Australian catchments (*Zhang & Post, 2018*).

Using simulated gaps to estimate runoff uncertainty does not require the statistical assumptions that parametric estimates require (e.g., homoscedasciscity, temporal independence). The gaps used to create the error distributions were simulated from the actual data, and thus represent the data structure better than estimates of error that assume theoretical error distributions. For example, confidence intervals surrounding runoff estimates are not required to be symmetrical, incorporating any biases present in the chosen infilling model into the resulting uncertainty estimate. This approach is similar to those commonly used to quantify uncertainty due to gaps in eddy flux measurements (*Richardson & Hollinger, 2007*) and can also be used to compare the efficacy of different gap filling models (*Falge et al., 2001*; *Harvey, Dixon & Hannaford, 2012*). Once error distributions have been created for each gap, it is easy to propagate the uncertainty in runoff due to gaps along with other sources of uncertainty using Monte Carlo sampling (*Campbell et al., 2016*; *Richardson & Hollinger, 2007*). The most appropriate infilling approach for a particular gap may differ depending on conditions (*Rees, 2008*), making the simulated gap approach particularly appealing, as it can easily provide errors associated with multiple models for use in Monte Carlo simulations.

One benefit of scrutinizing gaps in data sets is to evaluate options for reducing them. Maintenance (e.g., weir cleaning) is normally conducted when the error introduced by gap filling is smallest, at low flow. The single most important source of gaps was debris in the v-notch weir (44%), which has since been reduced by the addition of floating barriers in the ponding basin. The occurrence of gaps due to equipment malfunction (21%) has been

See et al. (2020), *PeerJ*, DOI 10.7717/peerj.9531

Peerj

**Table 2 Uncertainty in annual runoff due to gaps in watershed 6 at the Hubbard Brook Experimental Forest from 1996–2009.** Uncertainty is reported as the 95% CI (mm) and as a percentage of annual runoff in parentheses. In W6, 1 mm = 132,291 L. Confidence intervals < 0.1 mm are reported as 0.

| | 1996 | 1997 | 1998 | 1999 | 2000 | 2001 | 2002 | 2003 | 2004 | 2005 | 2006 | 2007 | 2008 | 2009 |
|---|---|---|---|---|---|---|---|---|---|---|---|---|---|---|
| Annual Flux | 601.3 | 699.8 | 496.9 | 757.9 | 847 | 950.7 | 897.7 | 871.8 | 828.8 | 742.8 | 1030.5 | 1403 | 808.7 | 1194.4 |
| All Gaps | 5.2 (0.4%) | 3.1 (0.5%) | 4.2 (0.4%) | 2.9 (0.4) | 10.0 (0.9%) | 8.2 (0.8%) | 1.8 (0.2%) | 9.9 (0.8%) | 1.9 (0.3%) | 13.7 (1.2%) | 1.5 (0.2%) | 16.2 (1.5%) | 1.6 (0.2%) | 21.1 (2.1%) |
| Debris in Weir | 0.45 (0.1%) | 1.7 (0.2%) | 1.4 (0.1%) | 1.0 (0.1%) | 0.3 (0%) | 8.3 (1.3%) | 1.7 (0.2%) | 0.4 (0.1%) | 1.0 (0.1%) | 0.2 (0%) | 0.2 (0%) | 0.6 (0.1%) | 0.1 (0%) | 2.7 (0.2%) |
| Maintenance | 0.2 (0%) | 0.9 (0.1%) | – | 0 (0%) | 0.1 (0%) | 0.2 (0%) | – | 2.0 (0.2%) | 0 (0%) | 0 (0%) | 0.1 (0%) | 0.1 (0%) | 0.1 (0%) | 0 (0%) |
| Equipment Failure | 0 (0%) | 2.2 (0.2%) | – | – | 9.8 (0.9%) | – | 0.8 (0.1%) | – | – | – | – | 16.4 (1.5%) | – | 21.3 (1.7%) |
| Human Error | 4.5 (0.6%) | – | 3.8 (0.3%) | – | – | – | – | 8.9 (1.2%) | – | – | 1.5 (0.1%) | – | 0.2 (0%) | – |
| Ice | 2.1 (0.2%) | 1.1 (0.1%) | 1.6 (0.1%) | 2.5 (0.4%) | – | 0.6 (0.1%) | 0.2 (0%) | 4.0 (0.4%) | 1.6 (0.2%) | 13.7 (1.2%) | 0.3 (0%) | 0.9 (0.1%) | 1.5 (0.2%) | – |

reduced by replacing antiquated mechanical chart recorders with digital sensors (optical encoders). The duration of gaps has been reduced by the radio transmission of electronic data, because problems can be identified and corrected more quickly than when weirs were visited weekly. Since many of these upgrades are recent, we expect that the number and duration of gaps has decreased compared with the time period reported here.

The rise of synthetic science and subsequent push for publicly available data sharing requires that data be properly curated and documented. Perhaps more than most environmental data, stream discharge estimates nearly always require a complete record due to their use in cumulative water or material flux calculations. The decision whether to archive data with gaps or with infilled values lies with the research team. When infilled values are included, they should be clearly identified as modeled values, so that data users can decide how to best treat them in their analyses. If gaps are filled, the infilling method should be described, along with the confidence associated with the modeled estimates (*Hamilton & Moore, 2012*).

## CONCLUSIONS

Computing advances in recent decades have allowed for a broader range of infilling techniques for streamflow data gaps, but the uncertainty associated with these new methods often cannot be assessed using traditional parametric methods. This work represents significant progress towards describing the uncertainty associated with infilling stream flow gaps in hydrologic datasets. Our estimates of uncertainty in runoff will contribute to uncertainty in estimates of other variables that rely on discharge, including stream solute loads (*Campbell et al., 2016*) and evapotranspiration (*Green et al., 2018*). Quantifying uncertainty provides the basis to prioritize improvements to streamflow monitoring strategies. In the hopes that others will benefit from and improve upon this method, we have made the code used in this analysis available in (Supplementary Information).

## ACKNOWLEDGEMENTS

We thank Brian Kronvang and Jamie Shanley for constructive feedback on an earlier version of this manuscript. Trevor Keenan suggested this approach to gap-filling during a Long Term Ecological Rresearch Synthesis Working Group meeting. This paper is a product of QUEST (Quantifying Uncertainty in Ecosystem Studies), http://www.quantifyinguncertainty.org and a contribution of the Hubbard Brook Ecosystem Study. Hubbard Brook is part of the LTER network supported by the National Science Foundation. The Hubbard Brook Experimental Forest is operated and maintained by the USDA Forest Service, Northern Research Station, Newtown Square, PA. Data analyzed in this paper are available at: doi:10.6073/pasta/282953c2290b1f00d9326ffd9a7e9668.

### Funding

This work was funded by the National Science Foundation DEB-1257906, DEB-1637685, and DEB-1114804. The funders had no role in study design, data collection and analysis, decision to publish, or preparation of the manuscript.

### Grant Disclosures

The following grant information was disclosed by the authors:
National Science Foundation: DEB-1257906, DEB-1637685, DEB-1114804.

### Competing Interests

The authors declare there are no competing interests.

### Author Contributions

- Craig R. See and Mark B. Green conceived and designed the experiments, performed the experiments, analyzed the data, prepared figures and/or tables, authored or reviewed drafts of the paper, and approved the final draft.
- Ruth D. Yanai and John L. Campbell conceived and designed the experiments, authored or reviewed drafts of the paper, and approved the final draft.
- Amey S. Bailey performed the experiments, analyzed the data, authored or reviewed drafts of the paper, and approved the final draft.
- Jeremy Hayward performed the experiments, authored or reviewed drafts of the paper, and approved the final draft.

### Data Availability

The code is available as a Supplemental File and the data are available at the Environmental Data Initiative: USDA Forest Service, Northern Research Station. 2019. Hubbard Brook Experimental Forest: Instantaneous Streamflow by Watershed, 1956 –present ver 12. Environmental Data Initiative.

https://doi.org/10.6073/pasta/282953c2290b1f00d9326ffd9a7e9668 (accessed 2020-06-25).

### Supplemental Information

Supplemental information for this article can be found online at http://dx.doi.org/10.7717/peerj.9531#supplemental-information.

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
