# Peer review of "Quantifying uncertainty in annual runoff due to missing data"

_PeerJ, doi:10.7717/peerj.9531_

## Round 0.1 · original submission · Minor Revisions

It is a good study and both reviewers asked a minor revision, which I agree. Please make a minor revision based on reviewers' comments.

·

Basic reporting

Dear Authors
I have with great pleasure been reviewing your manuscript entitled ‘Quantifying uncertainty in annual runoff due to missing data’ and find that your manuscript includes some very important scientific findings that are useful for all hydrologists and water quality scientists and managers.
Your language is very clear and a professional English language is used throughout the manuscript.
You have good and relevant references to the literature but I miss maybe references to the literature on uncertainty in load calculations for sediment and nutrients, which are very important for water managers. The uncertainty linked to e.g. daily flows during gap filling, might at high flows, be crucial for such load calculations.
Your structure conforms to PeerJ standards as far as I am aware and your Figures are relevant but could be improved for the readers to better see the results – e.g. Figure 4 and 5 – insert normal labelled axis – not logarithmic.
Raw data is also supplied as Supplementary Material.

Experimental design

Your results are clearly within the Scope of the journal. Moreover, your research questions are well defined and you have conducted a rigorous sampling and statistics to a high scientific standard. The methods used are described with sufficient detail and information to be replicated by others – e.g. for larger streams and rivers.

Validity of the findings

Your detailed analysis of gaps in hydrographs and impact of gap filling methods on runoff in a small stream is novel and give important inputs to hydrological science.
The only thing I am a little curious to know more about is the bias of runoff that you find in your assessments – as this might be of even greater concern than the uncertainty measured as the standard deviation in Figure 5. I would find that you should include an assessment of the bias in gap filling in your manuscript.

Additional comments

In conclusion, I find that you need to revise your very nice manuscript according to the few proposals mentioned above.

·

Basic reporting

A minor point, but Figures 2 and 3 units are inconsistent with the other figures.

Experimental design

I have noted in my general comments a few places where methods need clarification.

Validity of the findings

No comment.

Additional comments

This is a well-conceived study on a method for infilling high-frequency streamflow data for small catchments and estimating the associated uncertainty. The analytical approach is sound and reproducible, and the results are interpreted logically. I have only minor suggestions for improving the paper, by line number below.

Table 1. Why not indicate which of these 6 watersheds is the actual reference watershed where you are testing the infilling?

Line 38. "weir blockages" could be clarified and generalized to "debris on the control". Although weirs, with a restrictive geometric control, are highly prone to catch debris, most stream gages are not weirs.

71. Sentence beginning this line does not stand alone.

73. Capitalize “v” in V-notch as above.

77. “stage-discharge” is more conventional than “height-discharge”.

Figure 1 caption: I do not understand “in the model”, perhaps you mean “used in the model”? The left figure contains actual data, correct? Also, the inset technically shows all the data, not just the upper 2%.

95. Make clear the bin size is 1%. Maybe you intend this to be clear by “Percentile” but since you do discuss the 0.5 and 99.5% Qs, it could stand clarification.

95-97. Since the watershed sizes vary over a factor of 4, was there any thought about testing time-lagged regressions?

Figures 2 and 3. Units should also be in mm/hr for consistency and comparison to other plots.

128-130. I think you need to restate. I think you are performing 1000 sums, not summing 1000 values. But it needs further clarification – if you have 1000 error estimates for each gap and an average of 22 gaps per year it’s not clear how you come up with 1000 error estimates. And is there an averaging step?

133. It would be helpful to know the median gap length.

Figure 4. y-axis is runoff, not flow.

164-170. This paragraph would fit better as the next to last in the Discussion -after a more direct discussion of your results.

185. Homoscedasticity misspelled.

197. You could emphasize here, as you did in Intro, that stream discharge records often need to be complete; there is a stronger impetus for gapfilling than with most other environmental data series.

---

## Round 0.2 · accepted · Accept

The authors have responded well to review comments, so I am happy to accept it.